# Estimating individual risks of COVID-19-associated hospitalization and death using publicly available data

**Rajiv Bhatia**[1]*, **Jeffrey Klausner**[2]

**1** Department of Medicine (Affiliated), Stanford University, Stanford, California, United States of America,
**2** Department of Medicine and Public Health, University of California Los Angeles, Los Angeles, California, United States of America

* drajiv@stanford.edu

**Data Availability Statement:** All relevant data are within the manuscript and its Supporting information files except for COVID-19 case incidence data which can be accessed at the URL: https://github.com/nytimes/covid-19-data.

## Abstract

We describe a method to estimate individual risks of hospitalization and death attributable to non-household and household transmission of SARS-CoV-2 using available public data on confirmed-case incidence data along with estimates of the clinical fraction, timing of transmission, isolation adherence, secondary infection risks, contact rates, and case-hospitalization and case-fatality ratios. Using the method, we estimate that risks for a 90-day period at the median daily summertime U.S. county confirmed COVID-19 case incidence of 10.8 per 100,000 and pre-pandemic contact rates range from 0.4 to 8.9 per 100,000 for the four deciles of age between 20 and 60 years. The corresponding 90-day period risk of hospitalization ranges from 13.7 to 69.2 per 100,000. Assuming a non-household secondary infection risk of 4% and pre-pandemic contact rates, the share of transmissions attributable to household settings ranges from 73% to 78%. These estimates are sensitive to the parameter assumptions; nevertheless, they are comparable to the COVID-19 hospitalization and fatality rates observed over the time period. We conclude that individual risk of hospitalization and death from SARS-CoV-2 infection is calculable from publicly available data sources. Access to publicly reported infection incidence data by setting and other exposure characteristics along with setting specific estimates of secondary infection risk would allow for more precise individual risk estimation.

## Introduction

Perceptions of personal risk can modify pandemic disease transmission and population health outcomes by modifying behaviors such as mask wearing and by influencing demand for and compliance with government recommendations. A recent rapid review of studies that examined adherence with "quarantine" found adherence varied from 0% up to 92.8% and was influenced by social norms, perceived benefits of quarantine, perceived risk of the disease, and financial and material needs [1]. Public perceptions of risk also may influence how political leaders act. Ideally, public leaders should communicate risk precisely and transparently.

**Funding:** The author(s) received no specific funding for this work.

**Competing interests:** The authors have declared that no competing interests exist.

To be useful to individuals, estimates of risk need to be specific to people, times, places, and activities. Numerous reports have described clusters of SARS-CoV-2 infections in diverse settings, yet there has been little attention to estimating and communicating how personal risk varies by place, time, and population. Surveillance measures used to monitor and characterize the COVID-19 pandemic, including laboratory-based "case" counts of infection and mortality rates describe risk in aggregate. In the United States, public data on laboratory-confirmed infections, hospitalizations and deaths have not disaggregated information based on exposure factors even though case-reports include this information. No large-scale U.S. community transmission studies have comparatively evaluated risk by setting.

Recognizing the limitations of available data sources, we estimate and compare the person-level risks of SARS-CoV-2 infection associated hospitalization and death attributable to household and non-household contacts using public confirmed case incidence data and published transmission parameters. We identify knowledge gaps that, if filled, could make individual risk assessment more precise and useful.

## Materials and methods

We conceptualize risk as the product of susceptibility, infection incidence, the timing and efficiency of transmission from an infected individual, the contact setting, the contact rate, and the severity of infection. Because the true infection incidence rate is unknown, we assume a stable multiplicative relationship between the incidence rate of confirmed infections and the incidence of infections ($I \sim I_{Confirmed}$). We consider the transmission potential of clinical and subclinical infection separately, assuming that all symptomatic infected individuals will contribute to transmission during a pre-symptomatic infectious period and a share of symptomatic individuals will voluntarily self-isolate, effectively limiting transmission to within their own households. We assume those with subclinical infection will contribute to transmission for the duration of their infectious period but have a lower likelihood of forward transmission than those with symptoms.

Eq (1) estimates the age-specific probability of a confirmed infection attributable to contact within household settings, *P (Infection | HH)*$_i$, for age-strata $i$, as a function of the proportion susceptible to infection, $S$, the confirmed case incidence rate, $I$, the household specific secondary infection risk, $B_{HH}$, an age-specific household contact rate, $C_{HH\,i}$, the fraction of symptomatic infections, $F_{SX}$, and the complementary fraction of asymptomatic infections, *1- $F_{SX}$*, multiplied by an estimate of their relative infectiousness, r.

$$P\,(Infection\mid HH)_i = SIC_{HH\,i}\,B_{HH}\{F_{SX} + r(1 - F_{SX})\} \tag{1}$$

Eq (2) estimates the age-specific probability of a confirmed infection attributable to contact in non-household settings, *P (Infection | HH)*$_i$, for age-strata $i$. The share of pre-symptomatic transmission of the clinical fraction is denoted by p. The fraction of symptomatic infected cases that do not self-isolate, *1-q*, also contribute to transmission to non-household contacts during the period of post-symptomatic transmission.

$$P\,(Infection\mid NH)_i = SIC_{NH\,i}\,B_{NH}\{p\,(F_{SX}) + (1 - q)(1 - p)(F_{SX}) + r\,(1 - F_{SX})\} \tag{2}$$

We estimate the age-specific probabilities of hospitalization and death in Eqs (3) and (4) from the age specific probabilities of confirmed infection, multiplying these quantities by the

**Table 1. Parameters used for risk estimation.**

| Symbol | Parameter | Unit | Value | Source |
|---|---|---|---|---|
| $I$ | Average daily incidence rate of confirmed (reported) infections | Infections per day per person | Computed from data | Complied by the NYT from various sources |
| $S$ | Proportion of the population susceptible | Unitless proportion | 91% | Anand |
| $F_{SX}$ | Proportion of infections with clinical illness | Unitless proportion | 69% (95% CI: 63%–74%) | Buitrago-Garcia |
| $p$ | Percentage of transmission occurring prior to symptom onset | Unitless proportion | 50% | He, Casey, Ferrriti, US CDC |
| $q$ | Compliance with household self-isolation | Unitless proportion | 75% | Unsourced |
| $r$ | Relative Risk of infection transmitted from an asymptomatic individual relative to a symptomatic one | Unitless proportion | RR = 0.35 (95% CI, 0.10–1.27) | Buitrago-Garcia |
| $B_{HH}$ | Household Secondary infection risk | Infections observed per contact[a] during a monitoring period | 31.1% (95% CI 19.4% -42.7%) | Madewell |
| $B_{NH}$ | Non-household Secondary infection risk | Infections observed per contact during a monitoring period | 4.0% (95% CI: 2.8%, 5.2%) | Koh |
| $C_{iHH}$ | Household contact rate for age group $i$ | Contacts[b] / day | 20–29 years: 2.77 | Prem |
| | | | 30–39: 3.20 | |
| | | | 40–49: 3.18 | |
| | | | 50–59 3.33 | |
| $C_{iN_H}$ | Non-household contact rate for age group $i$ | Contacts / day | 20–29 y 11.88 | Prem |
| | | | 30–39 11.10 | |
| | | | 40–49 10.49 | |
| | | | 50–59 10.53 | |
| $CHR_i$ | Case Hospitalization ratio for age group $i$ | Unitless proportion | 20–29 years: 1.65% | US CDC surveillance case reports |
| | | | 30–39 years: 3.13% | |
| | | | 40–49 years: 4.75% | |
| | | | 50–59 years: 7.47% | |
| $CFR_i$ | Age-specific Case Fatality ratio for age group $i$ | Unitless proportion | 20–29 years: 0.05% | US CDC surveillance case reports |
| | | | 30–39 years: 0.15% | |
| | | | 40–49 years: 0.37% | |
| | | | 50–59 years: 0.97% | |

[a] Most studies estimating the SIR defined contact as but most commonly either as face to face unprotected contact for greater than 15 minutes, prolonged contact, or household membership

[b] The basis of these estimates, the POLYMOD study, defined contact as "...either skin-to-skin contact such as a kiss or handshake (a physical contact), or a two-way conversation with three or more words in the physical presence of another person but no skin-to-skin contact (a nonphysical contact).

age-specific confirmed case hospitalization and fatality ratios (CFRs), respectively.

$$P\left(Hospitalization\right)_i = CHR_i \times \{P\left(I|HH\right)_i + P\left(I|NH\right)_i\} \tag{3}$$

$$P\left(Fatality\right)_i = CFR_i \times \{P\left(I|HH\right)_i + P(I|NH)_i\} \tag{4}$$

We do not estimate risks for persons under 20 years due to the infrequent incidence of severe disease events in this age group. We also omit risk estimates for persons over 60 years as case hospital and fatality ratios cannot be reliably calculated for non-congregate settings using publicly available data. We list our parameters and their sources in Table 1 and discuss them further below.

We acquired data on confirmed SARS-CoV-2 infection incidence rates from publicly reported statistics compiled by The New York Times [2], computing average daily county-level confirmed infection incidence rates for the period of the "summer wave" in the U.S.

(June 16th to Sept 15th 2020), then finding the median incidence rate for each quintile of incidence.

Confirmed infection rates underestimate the true incidence of infection as not all people with symptomatic infections obtain tests, test produce false negative results and subclinical infections occur. The US CDC currently estimates that an average of 11 infections may occur for every confirmed case, but the source of this estimate indicates substantial regional variation [3]. We assume that confirmed infections and total infections will be proportional and related multiplicatively and that unconfirmed infections will have similar characteristics to confirmed ones. This assumption removes the need for application of this parameter in our methodology.

The true susceptible proportion of the population is also unknown. Pandemic risk assessments at the onset of the pandemic assumed a 100% susceptible population. Seroprevalence surveys conducted in US populations using convenience samples demonstrated low (<10%) antibody prevalence after the spring wave but substantial variability among regions [3]. A systematic study of the U.S. dialysis population estimated that seroprevalence of SARS-CoV-2 was 9·3% in US adult population, ranging from 3·5% in the west to 27·2% in the northeast [4]. Understanding the limitations of antibody-based determinations of susceptibility, we assumed the prevalence of susceptibility to be 91% based on the seroprevalence of Anand et al. but did not apply region-specific parameters in our methodology.

The parameters for secondary infection risk (SIR) (alternatively, secondary attack rates) come from reviews of community transmission studies. Studies included in reviews identify cases through active and passive surveillance and follow secondary contacts for a specified duration (typically 14 days), monitoring contacts for symptoms and, in most cases, testing contacts to confirm infection. Studies estimating the SIR define "contact" variously but commonly either as face to face unprotected (i.e., mask-less) contact for greater than 15 minutes, prolonged contact, or household membership. Studies compute the SIR as the number of infections that occur among the reported contacts of an ascertained case during the monitoring period. Studies typically do not ascertain the frequency of contact.

Published reviews of the SIR for SARS-CoV-2 have focused on household transmission. Koh et al reviewed 20 studies published by May 15, 2020, estimating a pooled household SIR at 15.4% (95% CI: 12.2%, 18.7%) [5]. The authors reported a higher SIR for adults relative to children (RR 1.40, 95% CI: 1.00, 1.96). This review also reported a summary non-household SIR of 4.0% (95% CI: 2.8%, 5.2%) which included several cluster investigations with high SIR estimates.

Lei et al. summarized 24 studies with data on household transmission risk published by July 1, 2020 including case reports with > 10 households, estimating a pooled SIR of 27% (95% CI, 21–32%) [6]. The review found a 3-fold higher risks for household transmission to adults. Data from ten studies with data on non-household contact, specifically, provided an SIR of 1.65% (95% CI, 1.43–1.87) based on 218 secondary cases among 13194 non-household contacts.

Madewell et al. estimated a summary SIR for 40 studies reporting SIRs for household and family contacts published through 31 July 2020, estimating a mean SIR for household contacts of 19·0% (95% CI, 14·9%–23·1%) and for family contacts of 18·1% (95% CI: 12·9%–34·8%) [7]. The estimated mean SIR for contact to adult household members was 31.1% (95% CI, 19.4% -42.7%). The authors also estimated a mean SIR for "close-contacts," which combined household and non-household contacts, as 4·3% (95% CI: 2·9%–5·6%).

For our risk estimates, we used the 31% summary estimate of the adult household SIR from Madewell et al. and the 4% summary estimate of the non-household SIR from Koh et al. We used the 1.65% estimate of the non-household SIR from Lei et al in a sensitivity analysis.

In the U.S., government agencies have not reported confirmed infection incidence by symptom status. Estimates of the clinical fraction from published studies have significant heterogeneity. A non-quantitative review of studies published from 19 April through 26 May 2020, reported asymptomatic fractions of 43–65% in community populations, 46–8% on cruise ships, 48–58% among personnel on aircraft carriers, 63–88% in the context of homeless shelter outbreaks, and 96% among inmates [8]. A meta-analysis of studies reported through 10 June estimated the asymptomatic fraction to be 20% (95% confidence interval [CI] 17–25) based on 79 studies with 6616 cases and 31% (95% CI 26%–37%) based on 7 studies with prospective follow-up [9]. Alternatively, a single large case-ascertained follow up study of 5,484 contacts which included both RT-PCR and serological examination, which limited under ascertainment from false negative RT-PCR tests, found that only 32% of test positive contacts developed symptoms of cough, shortness of breath with the clinical fraction varying from 18% for those under 20 to 65% for those over 80 years old [10]. Another meta-analysis of international studies also reported that clinical symptoms vary from 21% to 69% from the youngest to oldest subpopulations [11]. We used the 69% summary estimate of the clinical fraction reported by Buitrago-Garcia et al. from the subset of prospective studies and the lower 32% all-age-group clinical fraction reported by Polletti et al in a sensitivity analysis.

We assume that subclinical infections will contribute to non-household exposure for the duration of their infectious period while a fraction of those with clinical infections will self-isolate in their households following symptoms. We found no empirical data to estimate actual compliance with self-isolation requirements for the U.S. during the COVID-19 period. A recent survey of public attitudes in Israel found that willingness to comply with self-quarantine rose from 57% without governmental financial compensation to 94% with compensation [12]. We assumed that 75% of symptomatic individuals would self-isolate after developing symptoms and tested the sensitivity of our estimate to an alternative assumption of 50%.

Several transmission studies have estimated that roughly half of infection transmission occurs pre-symptomatically [13–15]. He et al. estimated that 44% (95% CI, 30–57%) of secondary cases were infected during the index cases' pre-symptomatic stage [13]. Based on a review of 17 studies reporting serial intervals or generation times, Casey et al. estimated that 56.1% of transmission occurred in the pre-symptomatic period [14]. Ferretti et al. found that the peak of transmission occurred at the time of symptom onset with 41% of transmission events occurring before symptoms onset and another 35% on the day of and day after symptom onset [15]. We estimate that 50% of infectiousness occurs before symptom onset.

Subclinical infections may be less infectious than clinical infections, because of differences in viral burden or the result of symptoms, such as coughing or sneezing. The review of Buitrago-Garcia reported that the secondary infection risk was lower (relative risk 0.35, 95% CI 0.10–1.27) among contacts of asymptomatic cases relative to symptomatic cases [9]. We considered infections in the subclinical fraction as being 35% as infectious as the clinical fraction. In a sensitivity analysis, we considered subclinical and clinical infections to be equally infectious.

We used age specific pre-pandemic estimates of contact rates from the work of Prem et al. who modeled age and location specific contact rates in 140 countries re-applying data from the European POLYMOD study [16]. The POLYMOD study defined contact as ". . .either skin-to-skin contact such as a kiss or handshake (a physical contact), or a two-way conversation with three or more words in the physical presence of another person but no skin-to-skin contact (a nonphysical contact)" [17]. This definition differs from the one applied most commonly for studies estimating the SIR for SARS-CoV-2. Prem et al. reported contact rates for home settings as well as for work, school and other settings. We summed the rates for non-

home settings as a non-household contact rate (S1 Table). As we were unable to obtain age-stratified case incidence data, we did not account for the age-structure of contacts.

We estimated the period case-hospitalization and case-fatality ratios by decile of age directly from US CDC surveillance case reports using the total counts of cases and those flagged for hospitalization or death [18]. We treated missing value in fields for hospitalization and death as non-hospitalized, non-fatal cases (S2 Table).

We computed risks for hospitalization and death for 90-day period using the average period confirmed case incidence and the parameters listed in Table 1. We utilized a Monte Carlo simulation to incorporate each equation parameter with an error ranges as a normally distributed variable, reporting risk as the mean of the resulting distribution.

To validate our estimates, we summed household and non-household risk and compared it to cumulative national age-specific mortality and hospitalization incidence rates for the same time period [19, 20] (S3 and S4 Tables).

## Results

Among US Counties, in the 90-day period from June 16th, 2020 to September 15th, 2020, the median daily confirmed case incidence was 10.8 per 100,000, varying by county from 3.1 per 100,000 in the first quartile to 28.5 per 100,000 in the fifth quartile.

Fig 1 illustrates the 90-day period risks for hospitalization and death attributable to household and non-household contacts across the range of U.S. county case incidence values applying the primary assumptions in Table 1.

Table 2 combines the household and non-household components of risk of hospitalization and death for each quintile of confirmed case incidence. The estimated 90-day period risk of death at the period overall median county confirmed case incidence of 10.8 per 100,000 ranges between 0.4 and 8.9 per 100,000 for each decile of age between 20 and 60 years. The corresponding 90-day period risk of hospitalization ranges from 13.7 to 69.2 per 100,000.

Though computed for different age-strata, our estimates are comparable to observed U.S. average age-specific mortality rates during the same time period of 0.48, 1.46, 4.39, 10.81, and 24.67 per 100,000, for 10-year age groups from 15–24 years to 55–64 years. Our estimates are also comparable to average age-specific hospitalization rates among the 98 counties in the

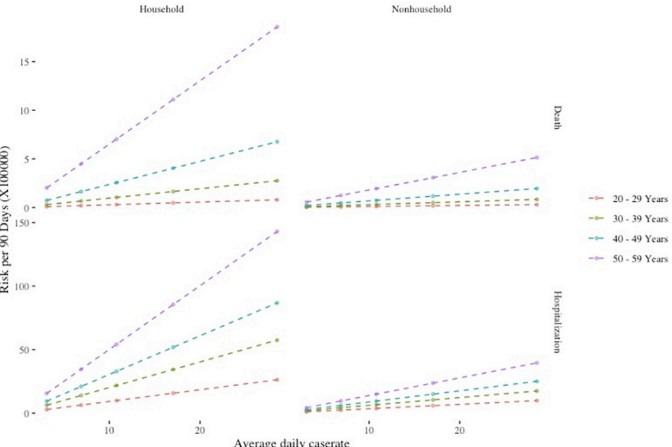

**Fig 1. Estimated risk of hospitalization and death over a 90-day period at pre-pandemic U.S. contact rates within the range of recent U.S. county case incidence under the following assumptions: Household SIR 31.1.%; non-household SIR, 4.0%; clinical fraction, 69%; relative infectiousness of asymptomatic infection, 35%; isolation adherence, 75%.**

**Table 2. Estimated risk of death and hospitalization at pre-pandemic U.S. contact rates over a 90-day time period at the median county incidence of confirmed infections during the "summer wave' of the COVID-19 pandemic (June 16 –September 15, 2020).**

| Quintile | Cumulative Period Risk of Death (X 100000) | | | | | Cumulative Period Risk of Hospitalization (X 100000) | | | | |
|---|---|---|---|---|---|---|---|---|---|---|
|  | 1 | 2 | 3 | 4 | 5 | 1 | 2 | 3 | 4 | 5 |
| Average Daily Case Incidence | 3.1 | 6.9 | 10.8 | 17.1 | 28.5 | 3.1 | 6.9 | 10.8 | 17.1 | 28.5 |
| 20–29 years | 0.1 | 0.3 | 0.4 | 0.7 | 1.1 | 4 | 8.8 | 13.7 | 21.6 | 36.2 |
| 30–39 years | 0.4 | 0.9 | 1.3 | 2.1 | 3.6 | 8.2 | 18.1 | 28.4 | 44.9 | 75 |
| 40–49 years | 0.9 | 2.1 | 3.3 | 5.2 | 8.8 | 12.1 | 27.1 | 42.4 | 66.9 | 111.9 |
| 50–59 years | 2.6 | 5.7 | 8.9 | 14.2 | 23.7 | 19.9 | 44.2 | 69.2 | 109.2 | 182.6 |

USCDC COVID-Net hospital surveillance program (18–29 years, 44.4 per 100,000; 30–39 years, 59.2 per 100,000; 40–49 years 78.5 per 100,000; 50–64 years 109.1 per 100,000).

Table 3 provides the 90-day period risk estimates disaggregated by setting of contact (household or non-household) at the national median county confirmed case incidence of 10.8 per 100,000 and illustrates the sensitivity of risk estimates to the alternative assumptions. Using the primary assumptions in Table 1 (scenario 1), for a person aged 40 to 49 years, the 90-day period risk of death attributable to household and non-household contacts is 2.56 and

**Table 3. Estimated risk of death and hospitalization at pre-pandemic U.S. contact rates over a 90-day time period at the median county incidence of confirmed infections during the "summer wave' of the COVID-19 pandemic (June 16 –September 15, 2020).**

| Age Group | Risk of Death (x 100000) | | Risk of Hospitalization (x 100000) | | Household Share |
|---|---|---|---|---|---|
|  | Household Contacts | Non-Household Contacts | Household Contacts | Non-Household Contacts |  |
| Scenario 1 (Base): Household SIR 31.1.%; Non-household SIR 4.0%, Clinical fraction 69%; Relative infectiousness of asymptomatic infection, 0.35; Isolation compliance 75% | | | | | |
| 20–29 Years | 0.3 | 0.11 | 9.96 | 3.74 | 73% |
| 30–39 Years | 1.04 | 0.32 | 21.78 | 6.62 | 77% |
| 40–49 Years | 2.56 | 0.74 | 32.89 | 9.5 | 78% |
| 50–59 Years | 7.04 | 1.95 | 54.18 | 15 | 78% |
| Scenario 2: Household SIR 31.1.%; Non-household SIR 1.65%, Clinical fraction 69%; Relative infectiousness of asymptomatic infection, 0.35; Isolation compliance 75% | | | | | |
| 20–29 Years | 0.3 | 0.05 | 9.89 | 1.55 | 86% |
| 30–39 Years | 1.04 | 0.13 | 21.62 | 2.75 | 89% |
| 40–49 Years | 2.54 | 0.31 | 32.65 | 3.95 | 89% |
| 50–59 Years | 6.98 | 0.81 | 53.77 | 6.23 | 90% |
| Scenario 3: Household SIR 31.1.%; Non-household SIR 4.0%%, Clinical fraction 31%; Relative infectiousness of asymptomatic infection, 0.35; Isolation compliance 75% | | | | | |
| 20–29 Years | 0.21 | 0.09 | 6.84 | 3.04 | 69% |
| 30–39 Years | 0.72 | 0.26 | 14.97 | 5.39 | 74% |
| 40–49 Years | 1.76 | 0.6 | 22.6 | 7.74 | 74% |
| 50–59 Years | 4.83 | 1.59 | 37.22 | 12.21 | 75% |
| Scenario 4: Household SIR, 31.1.%; Non-household SIR, 4.0%%, Clinical fraction, 69%; Relative infectiousness of asymptomatic infection, 1; Isolation compliance 75% | | | | | |
| 20–29 Years | 0.38 | 0.16 | 12.6 | 5.13 | 73% |
| 30–39 Years | 1.32 | 0.44 | 27.56 | 9.08 | 77% |
| 40–49 Years | 3.24 | 1.01 | 41.62 | 13.03 | 78% |
| 50–59 Years | 8.9 | 2.67 | 68.55 | 20.56 | 79% |
| Scenario 5: Household SIR, 31.1.%; Non-household SIR, 4.0%%, Clinical fraction, 69%; Relative infectiousness of asymptomatic infection, 0.35; Isolation compliance 50% | | | | | |
| 20–29 Years | 0.31 | 0.13 | 10.15 | 4.38 | 72% |
| 30–39 Years | 1.06 | 0.37 | 22.21 | 7.77 | 76% |
| 40–49 Years | 2.61 | 0.87 | 33.54 | 11.15 | 77% |
| 50–59 Years | 7.17 | 2.28 | 55.23 | 17.59 | 78% |

0.74 events in 100,000 respectively. For hospitalizations, the risk is 32.9 and 9.5 in 100,000 for household and non-household contacts, respectively. Assuming a non-household SIR of 4.0% and pre-pandemic contact rates, the share of transmissions attributable to household settings ranges from 73% to 78% depending on age.

Non-household risk falls proportionally using the lower estimate of the SIR of 1.65% (scenario 2). Decreasing the estimated clinical fraction to 31% (scenario 3) decreases both household and non-household risk estimates given we assume subclinical infections to be less infectious. In the second scenario, the relative size of the non-household share of risk also decreases slightly fewer infectious individuals are isolated. Assuming sub-clinical infections are equally infectious as clinical infections, (scenario 4) risks rise for both household and non-household contacts. Assuming adherence with isolation drops to 50% (scenario 5), risk increases for non-household contacts.

## Discussion

We demonstrate a straightforward method to estimate individual risks of COVID-19 associated hospitalization and deaths, using publicly available data on case incidence, the clinical fraction, transmission timing, secondary infection risk, contact rates, and case hospitalization and fatality ratios. These estimates of risk reflect the average risk across a wide range of exposure settings and do not account for individual risk factors for vulnerability to severe illness other than age. While many parameters have significant uncertainties, the comparability of estimated risks to observed hospitalization and fatality incidence rates validates the approach. Aggregated case incidence data, parameter uncertainty, and the lack of setting specific transmission risk estimates are limitations of the approach and suggest opportunities to improve individual risk estimation.

We assumed the susceptible fraction of the population remains high based on serological findings. The prevalence of detected antibodies varies with region and will change with time. Furthermore, protective immunity may not be well estimated by antibody detection alone. Durable lymphocyte responses may persist following exposure [21]. Observed cellular immune response to COVID-19 among unexposed individuals suggest prior exposure to related coronaviruses may further contribute to immunity [22].

Our estimates assume that prevalent infections are dispersed homogeneously within a county's geography. This does not account for clustering within chains of transmission among related or socially connected individuals. We might also expect higher infection incidence among those living in congregate living facilities, in neighborhoods with more density or larger households, or among service workers and those working together in close quarters. Disaggregated public reporting of confirmed infection incidence by age, symptom status, and neighborhood and congregate living status would allow for risk estimates to be setting and population specific.

Reviews of the secondary infection risk find significant heterogeneity among estimates. Most published estimates of the SIR come from observations outside the US and at an earlier time period in the pandemic, prior to normalization of behaviors intended to reduce the risk of infection transmission, such as increased hand washing, mask use and observing physical distance. SIR estimates are dependent on both contextual and study characteristics. Setting specific characteristics include the characteristics of human contact. Most studies defined contacts similarly, yet studies varied in the number of secondary contacts per primary case, suggesting variable interpretations of contact definitions or variable ability or wiliness to recall and disclose contacts.

Most studies used a combination of RT-PCR and symptom monitoring to identify infected contacts. RT-PCR methods will produce false negative results [23]. Studies that ascertained infection with RT-PCR only at the beginning of contact observation could have missed pre-clinical or subclinical infections.

The large difference between estimates of household and non-household SIR is not unexpected given that frequent, prolonged and intimate contact is the norm in households. Consistent with the relative size of the SIR estimates, we found that the majority of SARS-CoV-2 transmission can be attributed to households. The WHO mission to China also concluded that most infection transmission occurred in household settings [24]. An analysis of 1,038 cases identified in Hong Kong up to 28 April observed transmission to occur most frequently in households although non-household social settings could involve a larger number of secondary infections [25].

Relatively fewer studies have examined risk from non-household contact in detail; most of these occurred in East Asian countries and used data from governmental surveillance systems. We found insufficient estimates to apply SIRs for particular settings including workplaces, schools, and public transportation.

There has been little systematic characterization of contacts of confirmed cases of COVID-19 in the U.S. Reporting of the proportion confirmed cases with known contacts and the setting of contact or relationships among cases of contacts varies by jurisdiction; overall, few jurisdictions report this information. Among confirmed cases identified before 'stay-at-home' orders in nine Colorado counties participating in a retrospective survey, only 27% reported a contact with an infected individual [26]. Another survey found only 46% adults testing positive for SARS-CoV-2 infection could recall a contact with a known COVID-19 patient; of those who could recall a contact, most recalled contacts were family members (45%) or work colleagues (34%) [27].

Our estimates do not account for individual variation in the SIR. With respiratory viruses, the number of secondary cases generated by each index case can vary significantly [28]. Studies suggest that a large share COVID-19 infection might be due to a small fraction of particularly infectious individuals [25, 29]. Characteristics of COVID-19 individual "super-spreaders" are not described but these individuals may make a disproportionately larger contribution to the spread of infection in non-household, social settings [25]. Clusters of COVID-19 infections have occurred in diverse settings [30]. We applied the summary estimate of the non-household SIR by Koh et al which included estimates of the SIR from cluster investigations thus representing the potential contribution of "super-spreader" events.

We applied age-specific contact rates without regard to the age structure of contacts and based on pre-pandemic estimates. While changes in mobility and consumption suggest that many individuals may have lower contact rates, the nature and distribution of current social contacts is unknown. Furthermore, limits on social and workplace contacts outside the home may have increased household contact rates. Overall, we did not have a satisfactory way to adjust contact rates for post-pandemic conditions.

The definition of contact used for our contact rates, which includes physical touch or face to face speech, differs from contact definitions used in studies of secondary risk from SARS-CoV-2, which was usually defined as face to face "unprotected" contact for greater than 15 minutes. Neither definition may optimally characterize the mode of COVID-19 transmission. For example, neither definition explicitly consider transmission from contact via shared contaminated surfaces. Studies of the SIR that define contacts alternatively, for example, as physical work in close quarters, might produce useful findings.

We estimated the case hospitalization and fatality ratio by decile of age directly from line-level CDC surveillance case report data during the same period used to assess case-incidence.

We note that these ratios have varied significantly both with time and by location over the course of the pandemic; however, our examination of state-level ratios over time demonstrates that recent period estimates have more homogeneity (S1 Fig).

Our estimates do not account for variation in the risk of hospitalization and death due to individual co-morbid conditions [31]. Adjusting for these factors would lower estimated risks for most adults without common chronic disease conditions.

It is unclear how these risk estimates compare to of perceived individual risk among adults in the U.S as there are few published estimates of perceived risk. In one online survey conducted in March 2020, the median perceived risk was 10.0% for infection and 5.0% for infection fatality [32]. Another study conducted in March and April found that adults found that in March, 14% of U.S. adults perceived the fatality risk from SARS-CoV-2 infection to be greater than a 1% benchmark while 67% reported a lower risk; however, the perception of risk increased over time [33]. Daily media reports of counts of confirmed infections and the perceived lack of control over exposure all may be influencing risk perception.

Public risk perceptions may not be reflecting the relatively larger risk from household contacts. U.S. government action on restricting non-household social activity and the publicity surrounding disease clusters in social settings may have amplified perceived risks from non-household settings. In contrast, other countries took steps to quickly physically isolate infectious family member to limit transmissions from household contact [34, 35].

Return to community workplace and social life will require individuals to be have a better understanding of the personal risks of COVID-19 infection. Accurate setting and population specific estimates on the individual probabilities hospitalization and death may contribute to a more accurate risk perception. Systematically collected and publicly reported data on infection incidence by, for example, the geographic setting of exposure, residence type, whether a case had a known exposure, would allow more precise estimation than those possible with currently available public data. Calculation of secondary infection risks by setting and a more precise knowledge of susceptibility would improve individual risk estimates.

## Supporting information

**S1 Table. Daily contact rates by age and setting.**
(DOCX)

**S2 Table. Estimates of the case hospitalization and case fatality ratios.**
(DOCX)

**S3 Table. Cumulative COVID-19 associated hospital admissions rates per 100,000 people during the period June 16 to September 15, 2020.**
(DOCX)

**S4 Table. Cumulative COVID-19 associated mortality rates per 100,000 people during the period June 16 to September 15, 2020.**
(DOCX)

**S1 Fig. Estimates of the state-specific case fatality ratio by month.**
(DOCX)

## Author Contributions

**Conceptualization:** Rajiv Bhatia.

**Data curation:** Rajiv Bhatia.

**Formal analysis:** Rajiv Bhatia.

**Methodology:** Rajiv Bhatia, Jeffrey Klausner.

**Validation:** Rajiv Bhatia.

**Writing – original draft:** Rajiv Bhatia.

**Writing – review & editing:** Jeffrey Klausner.

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
