## [Decision Letter · Decision Letter 0]

14 Aug 2020

PONE-D-20-19975

Estimation of Individual Probabilities of COVID-19 Infection, Hospitalization, and Death From A County-level Contact of Unknown infection Status

PLOS ONE

Dear Dr. Bhatia,

Thank you for submitting your manuscript to PLOS ONE. After careful consideration, we feel that it has merit but does not fully meet PLOS ONE’s publication criteria as it currently stands. Therefore, we invite you to submit a revised version of the manuscript that addresses the points raised during the review process.

Please respond to the reviewer comments on a point-by-point basis and revise the manuscript accordingly.

We look forward to receiving your revised manuscript.

Kind regards,

Jeffrey Shaman

Academic Editor

PLOS ONE

Journal Requirements:

2.In your Data Availability statement, you have not specified where the minimal data set underlying the results described in your manuscript can be found. PLOS defines a study's minimal data set as the underlying data used to reach the conclusions drawn in the manuscript and any additional data required to replicate the reported study findings in their entirety. All PLOS journals require that the minimal data set be made fully available. For more information about our data policy, please see http://journals.plos.org/plosone/s/data-availability.

Reviewers' comments:

Reviewer's Responses to Questions

**Comments to the Author**

1. Is the manuscript technically sound, and do the data support the conclusions?

Reviewer #1: Yes

2. Has the statistical analysis been performed appropriately and rigorously? 

Reviewer #1: Yes

3. Have the authors made all data underlying the findings in their manuscript fully available?

Reviewer #1: Yes

4. Is the manuscript presented in an intelligible fashion and written in standard English?

Reviewer #1: Yes

5. Review Comments to the Author

Reviewer #1: # PONE-D-20-19975

## Overall comments

I would like to thank the editor for sending me an interesting paper to review with such a concise yet useful model for estimating the risk to residents of US cities of contracting SARS-CoV-2 and subsequent hospitalisation and death. Better informing the public about the risks of infection is vital to ensuring proportional public health responses and adherence to guidelines by the public.

Overall I am satisfied that the authors have thought through their work and its implications. The requested changes to the model above are meant to help the reader understand how much variability there is likely to be in the results based on uncertainty in published estimates used by the authors.

## Methods

In Table 1 there are no subscripts on the CHR and CFRs indicating which age group they belong to. Given that they are indeed age-dependent, it should be noted that equations (3) and (4) are age dependent. Additionally, the probability of an individual in any age group considered becoming infected is likely to be related to the total number of contacts that they have, the age group of those contacts (Prem et al., 2017), and the prevalence of current infection within each age group (although I understand this may be difficult to obtain). Contacts per day are given in the Results section (Table 3) but it is not clear in the Methods where these values are from.

Line 107: "can be both symptomatic and asymptomatic" should be "can be either symptomatic or asymptomatic".

Regarding the asymptomatic fraction, Buitrago-Garcia et al. (2020) provide a living systematic review of asymptomatic fraction that provides an estimate of 31% (95% prediction interval: 24%-38%), twice that which is used by the authors here. As asymptomatic infection may play an important role in ongoing transmission (e.g. Rivett et al., 2020) it would be worth the authors investigating the effect of a doubling of the asymptomatic fraction, particularly as asymptomatic infections will never have the chance to self-isolate on onset of symptoms.

Issues in case reporting are a major source of uncertainty in the estimation of prevalence and therefore risk of transmission. Russell et al. (2020) provide estimates of the reporting rate in the United States of America of 99% based on the method of Golding et al. (2020). The authors should be more explicit in how they came to the value of 75% which represents both under-reporting in symptomatic cases and undetected asymptomatic cases.

An assumption of 100% of infections having unknown contacts doesn't match the assumption of household transmission that the authors use. I appreciate that the authors have included this parameter in their model for future work. The discussion should highlight this as a limitation and it may be worth considering varying this assumption for sensitivity analysis.

In terms of culture and socioeconomics, I am skeptical that Israel provides a reasonable comparison to the United States of America for adherence to isolation guidelines. While financial compensation has been found to improve adherence there, an assumption that 75% of confirmed cases will voluntarily self-isolate is akin to assuming half of cases are having their income supported, given the numbers the authors provide. In contrast to income support in the United Kingdom and Singapore, income support in the United States of America is both low and spotty.

A review earlier this year (Webster et al., 2020) indicated that trust in government and other sociocultural factors may play a role, and while much of the studies cited are related to Ebola in Africa and H1N1 pandemic influenza in Australia, studies of SARS in Canada indicated that a sense of "civic duty" and a belief in the importance of "following the law" were associated with increased adherence. A recent survey indicates association between adherence and in a belief in a moral imperative to comply (van Rooij et al., 2020). There is evidence in the USA that a county's political demographics play a role in adherence (Painter and Qiu, 2020), particularly viewership of Fox News (Simonov et al., 2020), and poverty (Wright et al., 2020).

Given that the authors are considering county-level reported cases it may be worth estimating county-level rates of adherence to self-isolation guidelines.

## Results

I am satisfied, for the most part, with the presentation of results and their discussion. There are no uncertainties or sensitivity analyses presented in the results, due to the lack of uncertainty in the parameter estimates in the authors' model. I would suggest that estimates of incidence should have uncertainty due to uncertainty in under-reporting rates, CHR and CFR, and average days infectious - where the authors use 8 days without providing a source, despite indications that duration of infectivity may have a great deal of uncertainty due to differences in viral load (Wölfel et al., 2020).

Figure 1 is useful but I would also appreciate if the authors provided a figure with faceting by Event and colouring the lines by age group (colours to be consistent with the current Figure 2) in order to more easily see how the probability of each event varies with age group. The x axis is difficult to interpret, and the authors may want to consider converting from fractions to cases per 100,000.

## Discussion

The authors' discussion does a reasonable job of explaining the limitations of the data available to them and they compare their results appropriately.

At line 309 the authors the authors discuss mediation of risk perceptions based on restrictions on community action. I suggest they consider Webster et al. (2020) and their discussion on social norms and perception of risk.

## References

Prem et al. (2017) https://doi.org/10.1371/journal.pcbi.1005697

Rivett et al. (2020) https://doi.org/10.7554/eLife.58728

Russell et al. (2020) https://cmmid.github.io/topics/covid19/global_cfr_estimates.html

Golding et al. (2020) https://doi.org/10.1101/2020.07.07.20148460

Buitrago-Garcia et al. (2020) https://doi.org/10.1101/2020.04.25.20079103

Webster et al. (2020) https://doi.org/10.1016/j.puhe.2020.03.007

van Rooij et al. (2020) https://dx.doi.org/10.2139/ssrn.3582626

Painter and Qiu (2020) https://dx.doi.org/10.2139/ssrn.3569098

Simonov et al. (2020) https://doi.org/10.3386/w27237

Wright et al. (2020) https://dx.doi.org/10.2139/ssrn.3573637

Wölfel et al. (2020) https://doi.org/10.1038/s41586-020-2196-x

6. PLOS authors have the option to publish the peer review history of their article (what does this mean?). If published, this will include your full peer review and any attached files.

Reviewer #1: **Yes: **Samuel Clifford

---

## [Author Response · Author response to Decision Letter 0]

26 Oct 2020

Authors’ responses to reviewers’ comments

C1. I would like to thank the editor for sending me an interesting paper to review with such a concise yet useful model for estimating the risk to residents of US cities of contracting SARS-CoV-2 and subsequent hospitalization and death. Better informing the public about the risks of infection is vital to ensuring proportional public health responses and adherence to guidelines by the public.

R1. No response indicated

C2. Overall I am satisfied that the authors have thought through their work and its implications. The requested changes to the model above are meant to help the reader understand how much variability there is likely to be in the results based on uncertainty in published estimates used by the authors.

R2. No response indicated

C3. In Table 1 there are no subscripts on the CHR and CFRs indicating which age group they belong to. 

R3. We have specified the age-specific CHR and CFR estimates in the revision. 

C4. Given that they are indeed age-dependent, it should be noted that equations (3) and (4) are age dependent. 

R4. We have revised the description of formulae to note their age specificity.

C5. Additionally, the probability of an individual in any age group considered becoming infected is likely to be related to the total number of contacts that they have, the age group of those contacts (Prem et al., 2017), and the prevalence of current infection within each age group (although I understand this may be difficult to obtain). Contacts per day are given in the Results section (Table 3) but it is not clear in the Methods where these values are from.

R5. In the revision, we now use age and setting specific pre-pandemic contact rates from the analysis of Prem at al for our risk estimates. Given the lack of publicly available age-specific data on infection prevalence at the county level, we could not apply the age structure of contacts. We note the limitation and the potential value obtaining and applying age-structured incidence data in our revision

C6. Line 107: "can be both symptomatic and asymptomatic" should be "can be either symptomatic or asymptomatic".

R6. Corrected

C7. Regarding the asymptomatic fraction, Buitrago-Garcia et al. (2020) provide a living systematic review of asymptomatic fraction that provides an estimate of 31% (95% prediction interval: 24%-38%), twice that which is used by the authors here. As asymptomatic infection may play an important role in ongoing transmission (e.g. Rivett et al., 2020) it would be worth the authors investigating the effect of a doubling of the asymptomatic fraction, particularly as asymptomatic infections will never have the chance to self-isolate on onset of symptoms.

R7. The revision applies the estimate of the clinical fraction from Buitrago-Garcia et al and applies a lower estimate from the work of Poletti et al in a sensitivity analysis. 

C8. Issues in case reporting are a major source of uncertainty in the estimation of prevalence and therefore risk of transmission. Russell et al. (2020) provide estimates of the reporting rate in the United States of America of 99% based on the method of Golding et al. (2020). The authors should be more explicit in how they came to the value of 75% which represents both under-reporting in symptomatic cases and undetected asymptomatic cases.

R8. We have modified the methodology in the revision no longer apply or require a ratio between confirmed and unconfirmed infections. We have made and noted an assumption that the relationship between confirmed infections and all infections will be simply multiplicative within a time and place. 

C9. An assumption of 100% of infections having unknown contacts doesn't match the assumption of household transmission that the authors use. I appreciate that the authors have included this parameter in their model for future work. The discussion should highlight this as a limitation and it may be worth considering varying this assumption for sensitivity analysis.

R9. Changes to the methodology do not apply a parameter for unknown contacts. 

C10. In terms of culture and socioeconomics, I am skeptical that Israel provides a reasonable comparison to the United States of America for adherence to isolation guidelines. While financial compensation has been found to improve adherence there, an assumption that 75% of confirmed cases will voluntarily self-isolate is akin to assuming half of cases are having their income supported, given the numbers the authors provide. In contrast to income support in the United Kingdom and Singapore, income support in the United States of America is both low and spotty.

A review earlier this year (Webster et al., 2020) indicated that trust in government and other sociocultural factors may play a role, and while much of the studies cited are related to Ebola in Africa and H1N1 pandemic influenza in Australia, studies of SARS in Canada indicated that a sense of "civic duty" and a belief in the importance of "following the law" were associated with increased adherence. A recent survey indicates association between adherence and in a belief in a moral imperative to comply (van Rooij et al., 2020). There is evidence in the USA that a county's political demographics play a role in adherence (Painter and Qiu, 2020), particularly viewership of Fox News (Simonov et al., 2020), and poverty (Wright et al., 2020).

Given that the authors are considering county-level reported cases it may be worth estimating county-level rates of adherence to self-isolation guidelines.

R 10. We agree that data on adherence is insufficient. We were not able to find a reliable source for this parameter for the United States. We test the sensitivity of our estimates to this parameter.

C11. I am satisfied, for the most part, with the presentation of results and their discussion. There are no uncertainties or sensitivity analyses presented in the results, due to the lack of uncertainty in the parameter estimates in the authors' model. I would suggest that estimates of incidence should have uncertainty due to uncertainty in under-reporting rates, CHR and CFR, and average days infectious - where the authors use 8 days without providing a source, despite indications that duration of infectivity may have a great deal of uncertainty due to differences in viral load (Wölfel et al., 2020).

R11. We have added sensitivity analyses varying individual parameter estimates. We removed the parameter for duration of infectiousness as the SIR estimates include follow up duration that encompasses this interval.

C12. Figure 1 is useful but I would also appreciate if the authors provided a figure with faceting by Event and colouring the lines by age group (colours to be consistent with the current Figure 2) in order to more easily see how the probability of each event varies with age group. The x axis is difficult to interpret, and the authors may want to consider converting from fractions to cases per 100,000.

R12. The figure has been revised.

C13. The authors' discussion does a reasonable job of explaining the limitations of the data available to them and they compare their results appropriately.

R13. No response indicated

C14. At line 309 the authors the authors discuss mediation of risk perceptions based on restrictions on community action. I suggest they consider Webster et al. (2020) and their discussion on social norms and perception of risk.

R14. We reference the review of Webster in the introduction to the revision.

---

## [Decision Letter · Decision Letter 1]

16 Nov 2020

Estimating individual risks of COVID-19-associated hospitalization and death using publicly available data

PONE-D-20-19975R1

Dear Dr. Bhatia,

We’re pleased to inform you that your manuscript has been judged scientifically suitable for publication and will be formally accepted for publication once it meets all outstanding technical requirements.

Kind regards,

Jeffrey Shaman

Academic Editor

PLOS ONE

Additional Editor Comments (optional):

Reviewers' comments:

Reviewer's Responses to Questions

**Comments to the Author**

1. If the authors have adequately addressed your comments raised in a previous round of review and you feel that this manuscript is now acceptable for publication, you may indicate that here to bypass the “Comments to the Author” section, enter your conflict of interest statement in the “Confidential to Editor” section, and submit your "Accept" recommendation.

Reviewer #1: All comments have been addressed

2. Is the manuscript technically sound, and do the data support the conclusions?

Reviewer #1: Yes

3. Has the statistical analysis been performed appropriately and rigorously? 

Reviewer #1: Yes

4. Have the authors made all data underlying the findings in their manuscript fully available?

Reviewer #1: Yes

5. Is the manuscript presented in an intelligible fashion and written in standard English?

Reviewer #1: (No Response)

6. Review Comments to the Author

Reviewer #1: All my comments have been addressed and I have no further changes to request to the manuscript. Thank you.

7. PLOS authors have the option to publish the peer review history of their article (what does this mean?). If published, this will include your full peer review and any attached files.

Reviewer #1: **Yes: **Sam Clifford

---

## [Editor Report · Acceptance letter]

19 Nov 2020

PONE-D-20-19975R1 

Estimating individual risks of COVID-19-associated hospitalization and death using publicly available data 

Dear Dr. Bhatia:

I'm pleased to inform you that your manuscript has been deemed suitable for publication in PLOS ONE. Congratulations! Your manuscript is now with our production department. 

Kind regards, 

on behalf of

Prof. Jeffrey Shaman 

Academic Editor

PLOS ONE